# Association of inflammatory cytokines with obesity and pulmonary function testing

**Noor Al Khathlan**  *

Department of Respiratory Care, College of Applied Medical Sciences, Imam Abdulrahman Bin Faisal University, Dammam, Saudi Arabia

* naalkhathlan@iau.edu.sa

## Abstract

### Background

The World Health Organization (WHO) reported that the prevalence of obesity in the Kingdom of Saudi Arabia (KSA) is 33.7% (women 39.5% and men 29.5%), respectively. The effects of obesity on airway inflammation and respiratory mechanics as well as the function of adipose tissue has a key role in the development of various lung diseases. Therefore, this study aimed to compare the level of cytokines between obese (BMI $\geq$ 30) and non-obese participants and to assess their association with BMI, airways inflammation and pulmonary function.

### Method

One-hundred and seven non-smoking students (18–25 years of age) were recruited using convenience sampling technique for comparative cross-sectional study. Of them, 80 students were eligible and included in the analysis; 54 were non-obese (BMI<30) and 26 were obese (BMI $\geq$ 30). All the participants underwent anthropometric measurements, fractional exhaled nitric oxide (FeNO) measurement, spirometry and cytokines measurement (IL-6, IL-1β, GM-CSF, IL-7, IL-8 and IL-10). Measurements were compared between obese and non-obese groups. Then a correlation test was made between pro- and anti-inflammatory cytokines with BMI, pulmonary function test finding and FeNO.

### Results

The prevalence of obesity was 32.5% in the study population. Levels of pro-inflammatory cytokine IL-6 levels was significantly higher in obese than non-obese participants (p = 0.044). The level of FeNO log was significantly higher in obese participants than non-obese (p = 0.002). The pro-inflammatory cytokine IL-6 showed positive correlation with BMI while GMCSF showed negative correlation with FVC (p<0.05).

### Conclusion

The levels of pro-inflammatory cytokine IL-6 was found to be significantly higher in obese participants than non-obese participants. Furthermore, it showed positive correlation with BMI whereas pro-inflammatory cytokine GMCSF showed negative correlation with FVC.

**Data Availability Statement:** All relevant data are within the paper and its Supporting Information files.

**Funding:** NAK received fund from the Deanship of Scientific Research at Imam Abdulrahman Bin

Faisal University [grant number 2017-058-CAMS]. The funders had no role in study design, data collection and analysis, decision to publish, or preparation of the manuscript.

**Competing interests:** The authors have declared that no competing interests exist.

## Introduction

Obesity is a global health burden with over 1.9 billion adult individuals (39%) overweight, and over 650 million people (13%) obese as reported by epidemiological studies over 200 countries from 1975 to 2014 [1, 2]. According to the World Health Organization (WHO), the prevalence of overweight and obesity in the Kingdom of Saudi Arabia (KSA) is 68.2% and 33.7%, respectively [2]. Furthermore, a recent national survey conducted across all regions of Saudi Arabia revealed that the prevalence of obesity (BMI $\geq$ 30) was 24.7% [3]. Obesity is considered as a risk factor for developing respiratory symptoms and lung diseases, including exertional dyspnea, obstructive sleep apnea syndrome, obesity hypoventilation syndrome, chronic obstructive pulmonary disease, and asthma [3, 4]. This is because of the mechanical effects of obesity on lung physiology as well as the function of adipose tissue as an endocrine organ producing systemic inflammation [5].

It has been reported that adipocyte hypertrophy and hyperplasia in obese individuals may play an important role in inducing low-grade inflammation [6]. Expansion of adipose tissue and increased distance between adipocytes and capillaries results in hypoxic death of some adipocytes. In response to adipocyte death, pro-inflammatory macrophages M1 become active and remove debris from the damaged area [6]. During this process, adipocytes and M1 produce inflammatory cytokines including IL-6, TNF-α, IL-1β, and monocyte chemoattractant protein (MCP-1) [6].

The use of surrogate biomarkers blood eosinophils and fractioned exhaled nitric oxide (FeNO) combination provides the highest diagnostic accuracy for eosinophilic asthma. Eosinophils are normally present alongside the adipocytes and are suggestive of maintaining glucose balance and energy expenditure. The role of eosinophils in obesity is controversial as studies have reported conflicting results [7–9]. Moreover, obesity is associated with non-eosinophilic asthma. In neutrophilic asthma (non-eosinophilic) the dominant cytokines are IL-17, IL-21, and IL-22 [10, 11]. IL-17 is responsible for the staffing of neutrophils into the lungs [12]. Activation of the neutrophils takes place by the production of IL-6, granulocyte colony-stimulating factor (G-CSF), granulocyte-macrophage colony-stimulating factor (GM-CSF), IL-8, chemokine (C-X-C motif) ligand 1 (CXCL1), CXCL5, and CXCL8 from airway epithelial cells [10, 11].

Inflammasomes are also a possible way that translates obesity into chronic inflammation. NLRP3 is activated by saturated fatty acids like palmitate and stearate, free and crystal cholesterol and by oxidative stress which is present in adipose tissue in obesity [13]. During caloric excess, NLPR3 activation results in caspase-1 activation leading to inflammation [14].

Notably, there is only one national study that investigated the effects of overweight/obesity on lung function and airway inflammation and their relation to the level of Th1/Th2 cytokines [15, 16]. Therefore, we aimed in this study to compare the level of cytokines between obese (BMI $\geq$ 30) and non-obese participants and to assess their association with BMI, airways inflammation and pulmonary function.

## Subjects and methods

### Study design

This cross-sectional study was part of a research project evaluating the effect of obesity/overweight on airway inflammation and lung function and their association with the level of cytokines [15]. Participants were university students (18–25 years of age) recruited using the convenience sampling technique from different colleges at Imam Abdulrahman bin Faisal University. After reviewing the medical history of each participant, we excluded subjects with

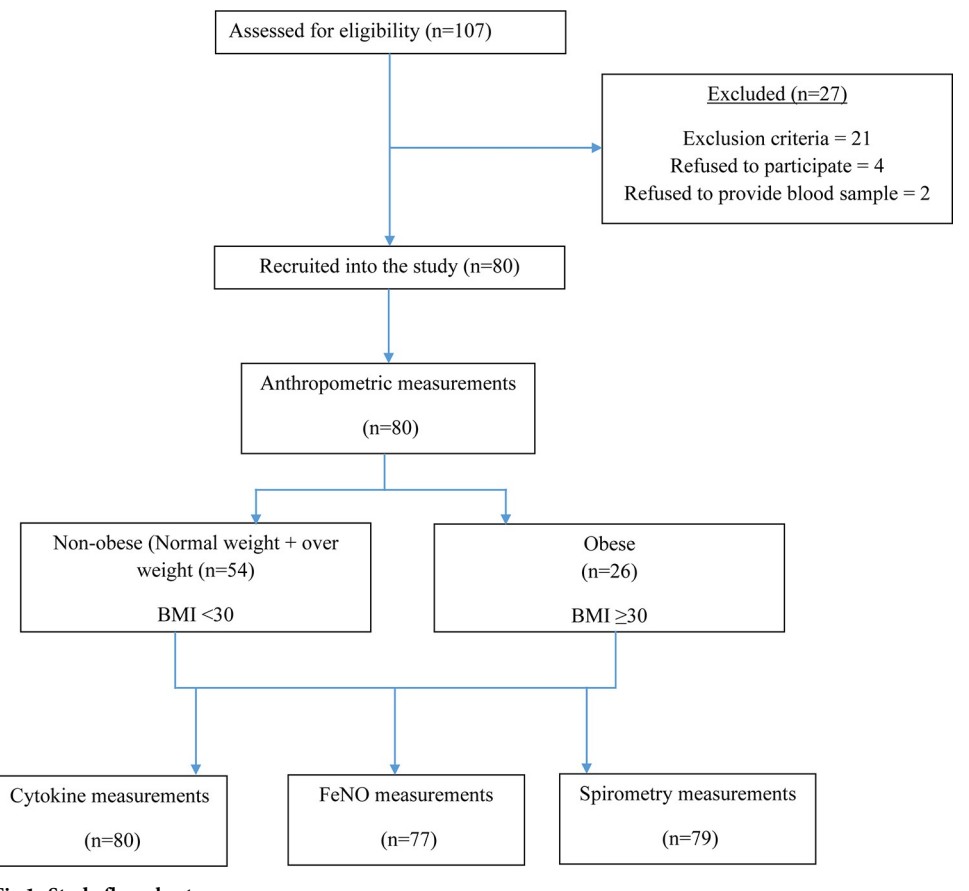

**Fig 1. Study flow chart.**

asthma, allergic rhinitis, chronic lung diseases, respiratory infection within 4 weeks of enrollment, pregnancy, and those who using bronchodilator or anti-inflammatory drugs within 7 days of enrollment. Therefore, from a total of 107 participants assessed for eligibility, 80 subjects were eligible and included in the final analysis; 54 were non-obese (BMI<30 kg/m$^2$) and 26 were obese (BMI $\geq$ 30 ~~30~~ kg/m$^2$). A flow diagram of the participants is illustrated in (Fig 1).

The study followed the principle of the Declaration of Helsinki 1995 (revised in 2013) [17]. Ethical approval was granted for the study by the institutional review board (IRB-2019-03-032, dated 29/1/2019) and a written consent was taken from the students before participation in the study.

**Test procedures.** All participants underwent the following tests in order: (1) anthropometric measurements, (2) fractional exhaled nitric oxide (FeNO) measurement, (3) spirometry and (4) cytokines measurement. FeNO measurement was done before spirometry to avoid any confounding effect of forced exhalation on the level of nitric oxide in the exhaled air.

- *Anthropometric measurements*: The participants were measured using a digital Secca electronic scale and Secca stadiometer for height and weight respectively with bare-foot and wearing light clothing. The measurements were adjusted to nearest 0.1 kg and 0.1 cm, respectively. The body mass index (BMI) was calculated as weight in kg/height in m$^2$.

- *Fractional Exhaled Nitric Oxide (FeNO) measurement*: The FeNO measurement was performed according to guidelines of the American Thoracic Society (ATS) [18] using a handled chemiluminescence NO analyzer (Niox Vero, Areocrine AB, Solna, Sweden). The mean

value of two exhalations at flow of 50 mL/s was used for the analysis. The participants were instructed not to exercise 1 hour prior to measurement and to avoid consumption of nitrate-rich foods (like sausage, spinach, cauliflower) 3 hours before the measurement [19].

- **Spirometry**: Spirometry was performed following ATS guidelines for standardization of spirometry using a calibrated computerized pneumatograph spirometer (Vitalograph) [20]. The absolute and percentage of predicted values of forced vital capacity (FVC), forced expiratory volume in the 1st second (FEV$_1$), forced expiratory flow at 25–75% (FEF$_{25-75\%}$), FEV$_1$/FVC ratio and peak expiratory flow (PEF) were calculated automatically based on age, sex, height and ethnicity, the predictive values for study participants were calculated based on Hankinson et al., reference equations [21]. Each participant performed at least three acceptable maneuvers and then the highest values were recorded.

- **Measurement of serum IL-6, IL-1β, GM-CSF, IL-7, IL-8 and IL-10 by ELISA**: The peripheral blood sample was collected from fasting participants in morning. Serum levels of the cytokines were determined by using commercially available ELISA kits (Human High Sensitivity T cell Panel Premixed 13-plexed- Immunology Multiplex Assay–Merck-Millipore).

**Statistical analysis.** Data were analyzed using Statistical Package of Social Science (IBM SPSS Statistics for Windows, Version 23, Armonk, NY). The continuous data are presented as mean ± standard deviation (SD). Unpaired *t*-test was used to compare FeNO levels and spirometry findings between the two groups as the data showed normal distribution, while the Mann-Whitney test was used to compare the levels of cytokines between the two groups as the data was not normally distributed. The correlation between BMI, cytokines level, FeNO and pulmonary test was performed using Spearman's correlation analysis. The categorical data are expressed as percentage and Pearson's chi-square test was used to compare the categorical data between the two groups. A p-value less than 0.05 is considered to be statistically significant.

## Results

This study was conducted on 80 university students who fulfilled the eligibility criteria for recruitment; 54 were non-obese (BMI<30 kg/m$^2$) and 26 were obese (BMI ≥ 30 30 kg/m$^2$). Out of these participants, 41 (51.25%) were in age group 18–20, and 41 (51.25%) were male participants. The prevalence of obesity was 32.5% (26/80) in the study population. No significant difference observed for age and gender between obese and non-obese participants (Table 1). The mean ± SD of IL 6, IL 1B, GMCSF, IL 7, IL 8 and IL 10 levels in the study participants were 6.12 ± 3.68 pg/ml, 0.64 ± 0.34 pg/ml, 8.44 ± 5.09 pg/ml, 14.19 ± 4.65 pg/ml, 6.13 ± 2.71 pg/ml and 3.34 ± 2.59 pg/ml, respectively.

**Table 1. Demographic and clinical characteristics of the study population.**

|  | Non-obese (n = 54) | Obese (n = 26) | p-value |
|---|---|---|---|
|  | n (%) | n (%) |  |
| Age (years) |  |  |  |
| 18–20 | 28 (51.9%) | 13 (50.0%) | 1.000 |
| >20 | 26 (48.1%) | 13 (50.0%) |  |
| Sex |  |  |  |
| Male | 26 (48.1%) | 15 (57.7%) | 0.480 |
| Female | 28 (51.9%) | 11 (42.3%) |  |

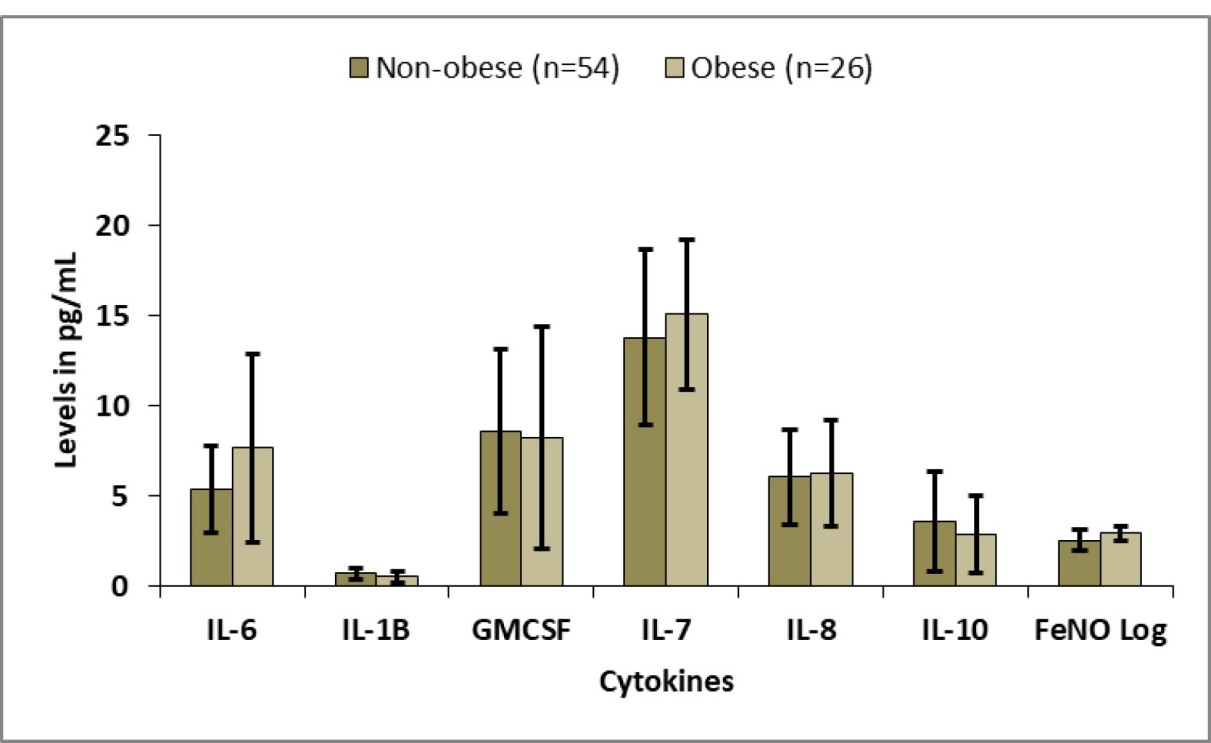

**Fig 2. Comparison of pro-inflammatory and anti-inflammatory cytokines between non-obese and obese group.**

## Comparison of cytokines levels between obese and non-obese participants

The levels of pro-inflammatory cytokine IL-6 was found to be significantly higher in obese participants than non-obese participants (7.64 ± 5.21 pg/ml vs 5.39 ± 2.40 pg/ml, p = 0.044). The levels of FeNO log was also found to be significantly higher in obese participants than non-obese participants (2.92 ± 0.38 pg/ml vs 2.55 ± 0.54 pg/ml, p = 0.002). The other cytokines showed no significant difference between non-obese and obese participants (Fig 2). The pulmonary test findings showed no significant difference between the non-obese and obese participants (p>0.05).

## Correlation between pro- and anti-inflammatory cytokines with BMI, pulmonary function test finding and FeNO

The pro-inflammatory cytokine IL-6 showed mild positive correlation with BMI ($r_s$ = 0.223, p = 0.046), while pro-inflammatory cytokine GMCSF showed mild negative correlation with FVC ($r_s$ = -0.239, p = 0.034). None of the other cytokines showed correlation with spirometric findings. Further, no association was observed between the cytokines level and FeNO log level (Table 2).

## Discussion

Saudi Arabia has become increasingly modernized, and now it has one of the highest obesity prevalence rates [22]. Obesity shows association with chronic systemic inflammation which arises due to the substance released by adipose tissue. Adipose tissue is an endocrine and energy storage organ which comprise of adipocytes, fibroblasts, endothelial cells and immune cells [23]. In our study we found that IL-6 levels were higher among the obese participants

**Table 2. Correlation between pro- and anti-inflammatory cytokines with BMI, pulmonary function test finding and FeNO.**

| | | IL-6 | IL-1B | GMCSF | IL-7 | IL-8 | IL-10 |
|---|---|---|---|---|---|---|---|
| BMI (kg/m$^2$) | $r_s$ | 0.223 | -0.215 | -0.105 | 0.032 | 0.054 | -0.128 |
| | p-value | 0.046* | 0.055 | 0.353 | 0.781 | 0.633 | 0.258 |
| FEV$_1$ (L) | $r_s$ | -0.013 | -0.143 | -0.194 | −0.206 | 0.052 | -0.135 |
| | p-value | 0.907 | 0.209 | 0.087 | 0.068 | 0.651 | 0.235 |
| FEV$_1$% of predicted | $r_s$ | 0.108 | -0.147 | -0.080 | -0.090 | 0.125 | -0.140 |
| | p-value | 0.344 | 0.197 | 0.484 | 0.430 | 0.274 | 0.219 |
| FVC (L) | $r_s$ | 0.085 | -0.110 | -0.239 | -0.219 | -0.044 | -0.149 |
| | p-value | 0.457 | 0.334 | 0.034* | 0.053 | 0.699 | 0.191 |
| FVC% of predicted | $r_s$ | 0.024 | -0.116 | -0.167 | -0.110 | 0.024 | -0.102 |
| | p-value | 0.832 | 0.310 | 0.141 | 0.334 | 0.831 | 0.370 |
| FEV$_1$/FVC | $r_s$ | 0.156 | -0.008 | 0.189 | 0.085 | 0.141 | 0.029 |
| | p-value | 0.170 | 0.944 | 0.095 | 0.454 | 0.216 | 0.800 |
| FEV$_1$/FVC predicted | $r_s$ | 0.068 | -0.080 | 0.053 | −0.010 | 0.123 | -0.052 |
| | p-value | 0.551 | 0.481 | 0.641 | 0.933 | 0.280 | 0.648 |
| PEF (L) | $r_s$ | -0.006 | -0.113 | 0.043 | -0.088 | 0.115 | -0.066 |
| | p-value | 0.956 | 0.322 | 0.704 | 0.944 | 0.313 | 0.564 |
| PEF% of predicted | $r_s$ | 0.085 | -0.048 | 0.173 | 0.031 | 0.083 | -0.052 |
| | p-value | 0.455 | 0.674 | 0.128 | 0.788 | 0.470 | 0.651 |
| FEF$_{25-75\%}$ (L/sec) | $r_s$ | -0.032 | -0.069 | -0.191 | -0.017 | 0.065 | -0.176 |
| | p-value | 0.779 | 0.546 | 0.094 | 0.883 | 0.570 | 0.124 |
| FEF$_{25-75\%}$ of predicted | $r_s$ | 0.024 | -0.093 | -0.145 | -0.053 | 0.061 | -0.096 |
| | p-value | 0.834 | 0.419 | 0.205 | 0.646 | 0.596 | 0.402 |
| FeNO Log (ppb) | $r_s$ | 0.024 | -0.004 | -0.019 | 0.135 | -0.158 | -0.097 |
| | p-value | 0.836 | 0.972 | 0.872 | 0.241 | 0.171 | 0.402 |

* p-value significant, $r_s$- Spearman's correlation coefficient

than non-obese participants. Further IL-6 levels showed positive correlation with BMI and thus it appears that IL-6 can function as a substitute marker for obesity.

The present findings were supported by another study findings which also reported IL-6 influences the transition from acute to chronic inflammation through stimulation of pro-inflammatory cytokines and downregulation of anti-inflammatory cytokines [24]. Similar findings have been reported by other researchers also that hypertrophic adipocytes shift the immune balance towards the production of pro-inflammatory molecules [25–29]. Studies have reported that shift in cytokine profile creates modification of white adipose tissue macrophage pool from activated M2 type (releasing anti-inflammatory cytokines IL 10, IL 1Ra, arginase) to classically-activated M1 type (releasing pro-inflammatory cytokines IL 6, TNF -alpha) [24–32]. Moreover, adiponectin are reduced in obesity which may be reason for increased release of pro-inflammatory cytokines like IL-6 [33].

Obesity can affect the respiratory system through a complex mechanism which can include direct mechanical change due to fat deposition as well as systemic inflammation. In our study we found a negative correlation of pro-inflammatory cytokine GMCSF level with FVC. Our study results are in concordance with the results of a study conducted on self-reported free from respiratory problem working men and women [34]. Another study reported that the pro-inflammatory cytokine IL-6 level has a negative correlation with pulmonary functions FEV$_1$% and FVC% [33].

In the present study spirometry findings showed no association with obesity, may be because the obese participants BMI were not too high ($35.24 \pm 4.31$ kg/m$^2$) and the population was below 30 years. Studies have reported that obesity cause stiffness of the lung and thoracic wall resulting in reduced respiratory compliance [35–37] but this occurs only in the condition when the individual is massively obese (BMI$\geq$40 kg/m$^2$) [38, 39].

FeNO is a biomarker that can help in identifying the inflammation in the airways [40]. In present study the level of FeNO was significantly higher in obese participants. Clinical studies have reported increased eosinophil airway's inflammation in obese asthmatic patients [41–43]. This may be a reason behind the severe course of asthma observed in presence of obesity [44].

## Limitation

The study has few limitations. The study was conducted among young adults from a single university. The generalizability of the results is reduced and small number participants with obesity might have hampered the statistical power of the findings.

## Conclusion

The levels of pro-inflammatory cytokine IL-6 was found to be significantly higher in obese participants than non-obese participants. Furthermore, it showed positive correlation with BMI whereas pro-inflammatory cytokine GMCSF showed negative correlation with FVC.

## Supporting information

**S1 Data.**
(XLSX)

## Author Contributions

**Conceptualization:** Noor Al Khathlan.

**Data curation:** Noor Al Khathlan.

**Formal analysis:** Noor Al Khathlan.

**Methodology:** Noor Al Khathlan.

**Resources:** Noor Al Khathlan.

**Writing – original draft:** Noor Al Khathlan.

**Writing – review & editing:** Noor Al Khathlan.

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
