## [Decision Letter · Decision Letter 0]

22 Aug 2023

PONE-D-23-15534Association of inflammatory cytokines with obesity and pulmonary function testingPLOS ONE

Dear Dr. Khathlan,

Thank you for submitting your manuscript to PLOS ONE. After careful consideration, we feel that it has merit but does not fully meet PLOS ONE’s publication criteria as it currently stands. Therefore, we invite you to submit a revised version of the manuscript that addresses the points raised during the review process.

We look forward to receiving your revised manuscript.

Kind regards,

Pisirai Ndarukwa, Ph.D.

Academic Editor

PLOS ONE

Journal Requirements:

https://lipidworld.biomedcentral.com/counter/pdf/10.1186/s12944-018-0778-5.pdf

http://www.bioline.org.br/request?pt06001=

https://mdpi-res.com/jcm/jcm-10-00169/article_deploy/jcm-10-00169.pdf?version=1609915862

In your revision ensure you cite all your sources (including your own works), and quote or rephrase any duplicated text outside the methods section. Further consideration is dependent on these concerns being addressed.

Additional Editor Comments:

Dear Author

I have read through this manuscript and is of interest. This paper will contribute significantly to science. However, it has to be substantially, reworked to meet the requirements for the journal if it can be accepted. The attached comments that came back from the reviewer must be addressed and should be addressed on a point to point format. These should be resubmitted together with a clean version of the manuscript to the journal for further considerations.

Regards

Reviewers' comments:

Reviewer's Responses to Questions

**Comments to the Author**

1. Is the manuscript technically sound, and do the data support the conclusions?

Reviewer #1: Partly

2. Has the statistical analysis been performed appropriately and rigorously? 

Reviewer #1: Yes

3. Have the authors made all data underlying the findings in their manuscript fully available?

Reviewer #1: Yes

4. Is the manuscript presented in an intelligible fashion and written in standard English?

Reviewer #1: Yes

5. Review Comments to the Author

Reviewer #1: This is a nice set of data. There is a need to improve the manuscript, especially the methods section for better understanding in the next review cycle.

There is great confusion while moving from the start towards the end of the manuscript due to insertion of ASQ and division into subgroups.

The basis on which the hypothesis was built is “a possibility of additive or synergistic effects between both inflammatory processes which may account for the reported association of asthma with obesity”. (Therefore, it was hypothesized that body mass index could influence the secretion of cytokines which in turn lead to increased airway inflammation.).

The reader thinks after reading the aforementioned lines that the study will make groups based on BMI and presence or severity of asthma

The next is the objective. “Hence, the present study was undertaken to compare the cytokines level between non-obese and obese participants and also to correlate the cytokines level with airways inflammation, pulmonary functions and asthma symptoms questionnaire.”

This affirms the opinion that the study is about correlations & comparisons between asthma patients of different BMI.

Now when we reach the methods, the thing that creates a confusion is that asthma is an exclusion criterion, at this stage the reader is in a fix about the real objective and hypothesis of this article.

“Exclusion criteria of the study were presence of any illnesses (asthma and allergic diseases, acute and chronic pulmonary diseases, acute or chronic infection, Cushing disease) and/or are on medication (use of steroids or anti-inflammatory medicines), which can modulate the cytokine secretion, and airways inflammation.”

Next as one reads through the methods, there is a confusing picture coming up about groups and subgroups. Although study population is identified, but the groups are not clearly mentioned.

It is not before the first paragraph of the results that one realizes that the groups were formed after including the subjects “just by chance”. It was entirely possible that there was no one with ASQ score >4. Likewise, they could be more or less than 26 and 54.

The Asthma Screening Questionnaire is basically used to separate asthmatics from non-asthmatics. The question is that if asthmatics are already excluded from the study then what is the use of classifying into score >4 or < 4?

“The obese and non-obese groups were further divided into 2 sub-groups based on the ASQ scores”

This confusion continues even in table-3 where suddenly numbers 1, 2, 3 and 4 appear for the four groups without any explanation (about which group is numbered what).

A mention about age based groups in the STROBE CHECKLIST “Age of students were grouped into two equal category18-20 years and 21-23 years for comparison” further adds to the confusion as this comparison is not visible in methods or results.

It could be a better study if there was a proper description of group formation in the methods. It is advisable to make a flow chart that should start from screening of 107, and reflect different stages passing through which the ultimate 80 were distributed.

A reminder about STROBE Guidelines

STROBE: 13 in results (a) Report numbers of individuals at each stage of study—eg numbers potentially eligible, examined for eligibility, confirmed eligible, included in the study, completing follow-up, and analysed

While c is Consider use of a flow diagram

In addition, the confusion about exclusion of asthma and then using ASQ at cut off of 4 and including both less than and more than 4 must be clarified.

“Asthma Screening Questionnaire score was grouped into two equal category 0-3 and 4-6 for comparison”. Surprisingly no reference is given for this questionnaire. Most probably the symptom based questionnaire by Shin et al was used, however reference must be given. (Shin B, Cole SL, Park SJ, Ledford DK, Lockey RF. A new symptom-based questionnaire for predicting the presence of asthma. J Investig Allergol Clin Immunol. 2010 Jan 1;20(1):27-34.)

Section Specific points (Most of them are already mentioned in the general section above)

Abstract:

“Therefore the present study was undertaken to compare cytokines level between nonobese

and obese participants along with their asthma screening questionnaire status. Further, the

study aimed to correlate cytokines level with body mass index, airways inflammation, and

pulmonary functions of the study participants.”

A reader cannot make up is the study addressing association of biomarkers with asthma, with obesity, asthma and obesity or obese asthmatics and nonobese asthmatics OR additive or synergistic effects.

Results of abstract should mention frequency of ASQ score (<4 or >4) in both the groups like obesity 32.6% is mentioned

Conclusion is very open ended with no link to actual objective (link of obesity with asthma markers) “High levels of IL-6 and FeNO showed association with obesity and this association existed in sub-group analysis between obese and non-obese participants with ASQ score <4, suggesting obesity as an independent factor affecting IL-6 and FeNO levels.”

Introduction

It is nicely written but too many references have been used.

Methods

The methods are very weak

There is no sample size mentioned in the methods although use of previous studies for sample size calculation is mentioned. What was the sample size that was determined, how the groups were formed, why the groups are unequal. Sample size should be applied to groups not to total.

One of the exclusion criteria is ‘asthma”, still the groups contain subjects who have ASQ scores suggestive of asthma. What is the reference for Asthma screening questionnaire (ASQ) and cutoff value of more or less than 4.

Results:

The first paragraph is part of methods not results (that too may be converted to a flow chart)

It is evident from second paragraph of results that distribution into nonobese and obese was not systematic, it was by chance that 26/80 were overweight or obese. It was possible that there were fewer, more or none. Likewise finding >4 ASQ score was also by chance, not planned (inclusion criteria)

Table-1: No statistically significant difference was observed between non-obese and obese participants for sex, age and ASQ score probably because asthmatics were excluded

Table-2: Flawed table. big confusion of p value, 1 vs 2, 2 vs 3 etc. What is 1,2,3,4, which of the subgroups? Why for IL6 and FeNO intergroup comparison is done and p value provided, while for the rest a single p value (probably by ANOVA) is provided. Please discuss with your statistician.

Discussion

Many paragraphs are not discussion, they are more suited for introduction. This has increased numbers of references tremendously. Please trim the discussion to just compare and contrast of this study’s results with the contemporary works and mechanisms wherever available.

“High levels of IL-6 and FeNO showed association with obesity and this association existed in sub-group analysis between obese and non-obese participants with ASQ score <4, suggesting obesity as an independent factor affecting IL-6 and FeNO levels.“ Without significant differenes, how could this be concluded?

Conclusion

It can be improved to reflect acceptance or rejection of hypothesis

6. PLOS authors have the option to publish the peer review history of their article (what does this mean?). If published, this will include your full peer review and any attached files.

Reviewer #1: No

---

## [Author Response · Author response to Decision Letter 0]

22 Oct 2023

Comment#1: There is a need to improve the manuscript, especially the methods section for better understanding in the next review cycle.

Reply: Methods section revised updated the inclusion criteria and added the flow chart describing the selection of participants.

Comment#2: There is great confusion while moving from the start towards the end of the manuscript due to insertion of ASQ and division into subgroups.

The basis on which the hypothesis was built is “a possibility of additive or synergistic effects between both inflammatory processes which may account for the reported association of asthma with obesity”. (Therefore, it was hypothesized that body mass index could influence the secretion of cytokines which in turn lead to increased airway inflammation.).

The reader thinks after reading the aforementioned lines that the study will make groups based on BMI and presence or severity of asthma

Reply: We have updated the objective and deleted ASQ in the revised manuscript.

Comment#3: The next is the objective. “Hence, the present study was undertaken to compare the cytokines level between non-obese and obese participants and also to correlate the cytokines level with airways inflammation, pulmonary functions and asthma symptoms questionnaire.”

This affirms the opinion that the study is about correlations & comparisons between asthma patients of different BMI.

Reply: We have updated the objective and deleted ASQ in the revised manuscript.

Comment#4: Now when we reach the methods, the thing that creates a confusion is that asthma is an exclusion criterion, at this stage the reader is in a fix about the real objective and hypothesis of this article.

“Exclusion criteria of the study were presence of any illnesses (asthma and allergic diseases, acute and chronic pulmonary diseases, acute or chronic infection, Cushing disease) and/or are on medication (use of steroids or anti-inflammatory medicines), which can modulate the cytokine secretion, and airways inflammation.”

Reply: We have updated the objective and deleted ASQ in the revised manuscript.

Comment#5: Next as one reads through the methods, there is a confusing picture coming up about groups and subgroups. Although study population is identified, but the groups are not clearly mentioned.

Reply: The subgroup is deleted as ASQ is deleted in revised manuscript.

Comment#6: It is not before the first paragraph of the results that one realizes that the groups were formed after including the subjects “just by chance”. It was entirely possible that there was no one with ASQ score >4. Likewise, they could be more or less than 26 and 54.

Reply: One-hundred and seven non-smoking students (18-25 years of age) were recruited using convenience sampling technique. Of them, 80 students were eligible and included in the analysis; 54 were non-obese (BMI<30) and 26 were obese (BMI ≥ 30). The groups were formed after we confirmed their eligibility. 

Comment#7: The Asthma Screening Questionnaire is basically used to separate asthmatics from non-asthmatics. The question is that if asthmatics are already excluded from the study then what is the use of classifying into score >4 or < 4?

Reply: ASQ is removed in the revised manuscript.

Comment#8: “The obese and non-obese groups were further divided into 2 sub-groups based on the ASQ scores”

This confusion continues even in table-3 where suddenly numbers 1, 2, 3 and 4 appear for the four groups without any explanation (about which group is numbered what).

Reply: Table 3 is deleted as ASQ is removed in the revised manuscript.

Comment#9: A mention about age based groups in the STROBE CHECKLIST “Age of students were grouped into two equal category18-20 years and 21-23 years for comparison” further adds to the confusion as this comparison is not visible in methods or results.

Reply: The age group is included in methods and also in results section (Table 1).

Comment#10: It could be a better study if there was a proper description of group formation in the methods. It is advisable to make a flow chart that should start from screening of 107, and reflect different stages passing through which the ultimate 80 were distributed.

A reminder about STROBE Guidelines

STROBE: 13 in results (a) Report numbers of individuals at each stage of study—eg numbers potentially eligible, examined for eligibility, confirmed eligible, included in the study, completing follow-up, and analysed

While c is Consider use of a flow diagram

Reply: Flow chart is included as Figure 1.

Comment#11: In addition, the confusion about exclusion of asthma and then using ASQ at cut off of 4 and including both less than and more than 4 must be clarified.

“Asthma Screening Questionnaire score was grouped into two equal category 0-3 and 4-6 for comparison”. Surprisingly no reference is given for this questionnaire. Most probably the symptom based questionnaire by Shin et al was used, however reference must be given. (Shin B, Cole SL, Park SJ, Ledford DK, Lockey RF. A new symptom-based questionnaire for predicting the presence of asthma. J Investig Allergol Clin Immunol. 2010 Jan 1;20(1):27-34.)

Reply: ASQ has been deleted in the revised manuscript. 

Section Specific points (Most of them are already mentioned in the general section above)

Abstract:

Comment#1: “Therefore the present study was undertaken to compare cytokines level between nonobese and obese participants along with their asthma screening questionnaire status. Further, the study aimed to correlate cytokines level with body mass index, airways inflammation, and pulmonary functions of the study participants.”

A reader cannot make up is the study addressing association of biomarkers with asthma, with obesity, asthma and obesity or obese asthmatics and nonobese asthmatics OR additive or synergistic effects.

Reply: The objective is restructured in revised manuscript.

Comment#2: Results of abstract should mention frequency of ASQ score (<4 or >4) in both the groups like obesity 32.6% is mentioned

Reply: ASQ is deleted in revised manuscript.

Comment#3: Conclusion is very open ended with no link to actual objective (link of obesity with asthma markers) “High levels of IL-6 and FeNO showed association with obesity and this association existed in sub-group analysis between obese and non-obese participants with ASQ score <4, suggesting obesity as an independent factor affecting IL-6 and FeNO levels.”

Reply: Conclusion updated as suggested in revised manuscript.

Introduction

Comment#1: It is nicely written but too many references have been used.

Reply: References has been reduced from 75 to 44.

Methods

Comment#1: The methods are very weak

There is no sample size mentioned in the methods although use of previous studies for sample size calculation is mentioned. What was the sample size that was determined, how the groups were formed, why the groups are unequal. Sample size should be applied to groups not to total.

Reply: The study participants are taken from main project and with these number of participants 2 articles have been published. 

International Journal of General Medicine 2020; 13: 955-962. 

Journal of Taibah University Medical Sciences 2022; 17 (1): 38-44. 

Comment#2: One of the exclusion criteria is ‘asthma”, still the groups contain subjects who have ASQ scores suggestive of asthma. What is the reference for Asthma screening questionnaire (ASQ) and cutoff value of more or less than 4.

Reply: ASQ is deleted in revised manuscript.

Results:

Comment#1: The first paragraph is part of methods not results (that too may be converted to a flow chart).

Reply: The first paragraph has been removed from the results section and Flow chart has been added in the revised manuscript.

Comment#2: It is evident from second paragraph of results that distribution into nonobese and obese was not systematic, it was by chance that 26/80 were overweight or obese. It was possible that there were fewer, more or none. Likewise finding >4 ASQ score was also by chance, not planned (inclusion criteria)

Reply: We recruited around 107 participants using convenience sampling technique. Of them, 80 students were eligible and included in the analysis; 54 were non-obese (BMI<30) and 26 were obese (BMI ≥ 30). The groups were formed after we confirmed their eligibility. 

Whereas ASQ has been deleted in the revised manuscript.

Comment#3: Table-1: No statistically significant difference was observed between non-obese and obese participants for sex, age and ASQ score probably because asthmatics were excluded.

Reply: Yes, we accept these findings may be due to exclusion of asthmatics.

Comment#4: Table-2: Flawed table. big confusion of p value, 1 vs 2, 2 vs 3 etc. What is 1,2,3,4, which of the subgroups? Why for IL6 and FeNO intergroup comparison is done and p value provided, while for the rest a single p value (probably by ANOVA) is provided. Please discuss with your statistician.

Reply: Table 2 is deleted as the ASQ has been removed in the revised manuscript.

Discussion

Comment#1: Many paragraphs are not discussion, they are more suited for introduction. This has increased numbers of references tremendously. Please trim the discussion to just compare and contrast of this study’s results with the contemporary works and mechanisms wherever available.

Reply: The discussion has been revised as suggested and references have been reduced from 75 to 44.

Comment#2: “High levels of IL-6 and FeNO showed association with obesity and this association existed in sub-group analysis between obese and non-obese participants with ASQ score <4, suggesting obesity as an independent factor affecting IL-6 and FeNO levels.“ Without significant differences, how could this be concluded?

Reply: Conclusion has been revised as ASQ is removed in the revised manuscript.

Conclusion

Comment#1: It can be improved to reflect acceptance or rejection of hypothesis

Reply: Conclusion has been revised as suggested.

---

## [Decision Letter · Decision Letter 1]

6 Nov 2023

Association of inflammatory cytokines with obesity and pulmonary function testing

PONE-D-23-15534R1

Dear Dr. Al Khathlan,

We’re pleased to inform you that your manuscript has been judged scientifically suitable for publication and will be formally accepted for publication once it meets all outstanding technical requirements.

Kind regards,

Pisirai Ndarukwa, Ph.D.

Academic Editor

PLOS ONE

Additional Editor Comments (optional):

Reviewers' comments:

Reviewer's Responses to Questions

**Comments to the Author**

1. If the authors have adequately addressed your comments raised in a previous round of review and you feel that this manuscript is now acceptable for publication, you may indicate that here to bypass the “Comments to the Author” section, enter your conflict of interest statement in the “Confidential to Editor” section, and submit your "Accept" recommendation.

Reviewer #1: All comments have been addressed

2. Is the manuscript technically sound, and do the data support the conclusions?

Reviewer #1: Partly

3. Has the statistical analysis been performed appropriately and rigorously? 

Reviewer #1: Yes

4. Have the authors made all data underlying the findings in their manuscript fully available?

Reviewer #1: Yes

5. Is the manuscript presented in an intelligible fashion and written in standard English?

Reviewer #1: No

6. Review Comments to the Author

Reviewer #1: (No Response)

7. PLOS authors have the option to publish the peer review history of their article (what does this mean?). If published, this will include your full peer review and any attached files.

Reviewer #1: No

---

## [Editor Report · Acceptance letter]

13 Nov 2023

PONE-D-23-15534R1 

Association of inflammatory cytokines with obesity and pulmonary function testing 

Dear Dr. Al Khathlan:

I'm pleased to inform you that your manuscript has been deemed suitable for publication in PLOS ONE. Congratulations! Your manuscript is now with our production department. 

Kind regards, 

on behalf of

Dr. Pisirai Ndarukwa 

Academic Editor

PLOS ONE